# Design and 3D Printing of Stretchable Conductor with High Dynamic Stability

**DOI:** 10.3390/ma16083098

**Published:** 2023-04-14

**Authors:** Chao Liu, Yuwei Wang, Shengding Wang, Xiangling Xia, Huiyun Xiao, Jinyun Liu, Siqi Hu, Xiaohui Yi, Yiwei Liu, Yuanzhao Wu, Jie Shang, Run-Wei Li

**Affiliations:** 1School of Materials Science and Chemical Engineering, Ningbo University, Ningbo 315211, China; 2CAS Key Laboratory of Magnetic Materials and Devices, Ningbo Institute of Materials Technology and Engineering, Chinese Academy of Sciences, Ningbo 315201, China; 3Zhejiang Province Key Laboratory of Magnetic Materials and Application Technology, Ningbo Institute of Materials Technology and Engineering, Chinese Academy of Sciences, Ningbo 315201, China; 4College of Materials Science and Opto-Electronic Technology, University of Chinese Academy of Sciences, Beijing 100049, China

**Keywords:** liquid metal, stretchable conductor, 3D printing, high dynamic stability, wearable devices

## Abstract

As an indispensable part of wearable devices and mechanical arms, stretchable conductors have received extensive attention in recent years. The design of a high-dynamic-stability, stretchable conductor is the key technology to ensure the normal transmission of electrical signals and electrical energy of wearable devices under large mechanical deformation, which has always been an important research topic domestically and abroad. In this paper, a stretchable conductor with a linear bunch structure is designed and prepared by combining numerical modeling and simulation with 3D printing technology. The stretchable conductor consists of a 3D-printed bunch-structured equiwall elastic insulating resin tube and internally filled free-deformable liquid metal. This conductor has a very high conductivity exceeding 10^4^ S cm^−1^, good stretchability with an elongation at break exceeding 50%, and great tensile stability, with a relative change in resistance of only about 1% at 50% tensile strain. Finally, this paper demonstrates it as a headphone cable (transmitting electrical signals) and a mobile phone charging wire (transmitting electrical energy), which proves its good mechanical and electrical properties and shows good application potential.

## 1. Introduction

Flexible electronic devices have mechanical properties that traditional devices do not have, such as bendability [1,2,3], foldability [4,5,6] and stretchability [7,8,9], and are widely used in healthcare [10,11], medicine [12,13], and soft robots [14,15,16]. Stretchable conductors, as the basic components of flexible electronic devices, have always been a hot topic of research. Compared with traditional rigid conductors, stretchable conductors have more deformability, which greatly expands the applications of wearable devices, and saves a lot of wiring space for robotic arms [17]. Stretchable conductors maintain stable conductivity under large deformation, that is, high dynamic stability, which is crucial for the stable operation of devices such as robotic arms and wearable electronics devices. In order to fully reflect the advantages of stretchable conductors, it is necessary to have both high stretchability and stable conductivity [18].

As a key component of flexible electronic devices, stretchable conductors are usually realized by the structure or material [19,20,21,22,23,24]. Structure-based stretchable conductors are one of the most widely used methods to achieve scalable interconnections. Some non-stretchable conductive materials are designed as serpentine [25], mesh [26,27], cracks [28,29,30], and longitudinal waves [31,32]. However, high stretchability usually requires more complex patterns, which increases the wire resistance and process complexity. In terms of materials, conductive composites are mainly prepared by using polymers that are inherently conductive or mixing conductive materials into elastomers. For example, poly (3,4-ethyldioxythiophene):poly (styrenesulfonate) (PEDOT:PSS) is intrinsically conductive. The modified PEDOT:PSS film has more than 50% mechanical stretchability and more than 1000 S cm^−1^ of conductivity [33]. Despite this, intrinsically conductive polymers have disadvantages such as high cost and poor stability [34]. In addition, conductive networks can also be provided by mixing conductive materials in elastic polymers, such as metal nanowires [35,36], graphene [37,38], carbon nanotubes [39,40], and liquid metals [41]. This polymer composite material allows the manufacture of flexible and scalable conductors through a simple mixing process. However, it is difficult for this conductive composite material to maintain a stable resistance during deformation, and in repeated deformation, due to the mutual extrusion of conductive materials or the disconnection of the conductive path. These shortcomings limit the practical application of conductive composites as stretchable conductors [42].

Some researchers have achieved relatively stable resistance changes during stretching by constructing a three-dimensional network of conductive materials. For example, Gao et al. reported a stretchable conductor inspired by the maple leaf shape. A prestrain finishing method was used to make the surface of the conductive polypyrrole coating a layered wrinkle structure, so the resistance change was 66% under 600% stretching and its conductivity was 100 S m^−1^ [43]. Zhang et al. formed a three-dimensional conductive network by welding carbon nanotubes, and then packaged it with PDMS. The resistance change was about 5% and the conductivity was about 132 S m^−1^ under 90% stretching [44]. Hong et al. used the bimodal porous structure made of silver nanowires. At 40% strain, the resistance change was 8% and the conductivity was 42 S cm^−1^ [45]. Nevertheless, most of the reported stretchable conductors have electrical degradation or a conductivity that cannot meet the requirements of practical applications, and the preparation process is complicated so cannot be prepared in batches.

There are other examples of stretchable conductors by using intrinsically flexible liquid metal. Liang et al. filled liquid metal into a porous sponge made of elastomers, the maximum conductivity could reach 10,000 S cm^−1^, and the resistance change was about 10% at 50% stretch, but it required a large liquid metal consumption and was not easy to package [46]. Ning et al. improved its mechanical properties by mixing liquid metal and elastic matrix PUS and using PDMS for encapsulation. The conductivity was 478 S cm^−1^, and the resistance change was about 2% at 50% stretch [47]. This highly loaded composite material is prone to liquid metal leakage after multiple stretching and the comparably low conductivity is also problematic. Injecting liquid metal into the elastomer channel also makes it possible to prepare high-conductivity stretchable conductors, but the complex preparation process and the great resistance change after stretching are problematic. It is also difficult to obtain high conductivity by mixing with materials, because it is difficult for micron-sized liquid metal droplets with low aspect ratio to form a dense conductive network.

In this paper, to develop the next generation of wearable flexible electronic devices and improve the efficiency of expensive liquid metals, an innovative linear bunch three-dimensional conductive structure is proposed to achieve stable resistance change within a certain strain range. Stretchable conductors with high dynamic stability are fabricated by designing strain-insensitive elastomeric channels that can be prepared in batches through the 3D printing technology developed in recent years. Liquid metal is selected as the conductive filler of a three-dimensional conductive structure. Because of its inherent deformability and good conductivity, it can respond to external stress and undergo reversible deformation without any hysteresis or mechanical degradation, which avoids the phenomenon of electrical degradation during repeated stretching. The results show that the liquid-metal-based stretchable conductor with a linear bunch three-dimensional conductive network has an excellent conductivity of up to 10,000 S cm^−1^, and the elongation at break is greater than 50%. More importantly, it has an excellent dynamic stability during stretching. At 50% tensile strain, the relative resistance change (ΔR/R_0_) is only about 1%, which shows great prospects as a high-performance stretchable conductor and is of great significance to the development of flexible electronic devices. In subsequent practical application testing, as an elastic headphone line, it has an excellent electrical signal transmission capability and has little effect on the music signal after stretching 50%. As an elastic charging line for mobile phones, it still transmits power stably under dynamic stretching.

## 2. Materials and Methods

### 2.1. Materials

Photocurable resin Agilus30 (Stratasys, Eden Prairie, MN, USA), high-purity metal gallium (99.99%; Beijing Founde Star Sci. & Technol. Co., Ltd., Beijing, China), and indium (99.995%; Beijing Founde Star Sci. & Technol. Co., Ltd.) were used. The extra reagents utilized in the experiment were altogether gained from Sinopharm Chemical Reagent Co., Ltd., Shanghai, China.

### 2.2. Preparation Process

The linear bunch structure was fabricated using a photocuring 3D printer (AUTOCERA-L, Beijing Shiwei Technology Co., Ltd., Beijing, China). The elastic resin Agilus30 was used for molding, the molded sample was then placed in a beaker filled with alcohol, and the ultrasonic machine was used to ultrasound at 25 kHz and 25 °C for 10 min. The object after cleaning was put into the oven for drying for 5 min, and then the sample was post-cured for 20 min by a UV curing machine. After curing completely, the bracket was removed. We mixed gallium and indium in a mass ratio of 3:1, then heated the mixture in a water bath at 60 °C and stirred for 30 min to obtain a liquid metal (gallium indium alloy). A syringe was used to fill the liquid metal with a bunch-structured conductive network, and the elastic resin Agilus30 was used at both ends to fix the copper sheet and seal the port for easy testing.

### 2.3. Young’s Modulus

The Young’s modulus of the elastic resin Agilus30 was determined by tensile testing. Standard stretch parts were made using a 3D printer, and the sample was secured on a stretching machine (Instron 5943, Norwood, MA, USA) with an initial length of 2 cm. The sample was stretched at a rate of 50 mm min^−1^, and the obtained stress–strain curve was linearly fitted to confirm that the slope was Young’s modulus.

### 2.4. Computational Simulations

COMSOL Multiphysics 6.0 was used for finite element simulation. The Yeoh model in hyperelastic materials was used for simulation experiments, the measured material parameters were input, and the density of the material was set to 970 kg m^−3^. As liquid metal is fluid, it is necessary to adopt a multi-physical field of fluid–solid coupling. In the simulation, one end of the sample was fixed and the other end was stretched. As the stretching proceeded, the shape of the sample changed. The electrical module recorded the change in resistance according to the formula R = ρL S^−1^ of resistance.

### 2.5. Mechanical Features

The basic mechanical properties such as the strain and strength of the printed linear-bunch-structured stretchable conductor were tested by a stretcher (Instron 5943).

### 2.6. Electrical Characteristics

The resistance was measured using the DC current source (Keithley 6221, Cleveland, OH, USA) and the nanovoltmeter (Agilent 34420A, Santa Clara, CA, USA) by the four-wire method. The copper sheets at both ends of the sample were connected to the current source and voltmeter, and the data acquisition system was used to observe the resistance change of the linear-bunch-structured stretchable conductor under the stretching of the stretching machine (Instron 5943).

### 2.7. Making a Headphone Cable

Part of the earphone cable was replaced with the prepared sample so that the two ends of the earphone wire were bonded to both ends of the sample by light-curing resin, and a stretchable earphone cable was obtained. To test the voltage waveform, an oscilloscope (710 110/DLM2024, Yokogawa, Tokyo, Japan) was connected in parallel with a speaker to record the music signal.

## 3. Results

### 3.1. Structural Design

Figure 1a shows the three strain-insensitive structures designed in this paper, and the three shapes are numbered as 1, 2, and 3. When the structure with uneven thickness is stretched, the deformation squeezes the liquid metal in the coarse position so that the fine position is supplemented, which may retain the overall resistance in a relatively stable state during the stretching process. Taking shape 2 as an example, the structural parameters that control its shape are analyzed. The coarse local radius is set to the outer diameter, the fine position is set to the inner diameter, and the distance between the two adjacent inner diameters is set to the length, as shown in Figure 1b. Then, a comparative analysis of the designed structures is conducted to select the most strain-insensitive structure for subsequent analysis. Therefore, an orthogonal test table (Appendix A) with four factors and three levels is designed. The shape and structural parameters of the outer diameter, inner diameter, and length are listed as the factors affecting the relative resistance change after stretching. Each factor is given three values. The relative resistance change of the stretchable conductor after 50% stretching in the orthogonal experiment is calculated by the finite element method. Figure 1c shows the estimated marginal mean of each factor. Its role is to control the results of each independent variable to predict the average of the dependent variables when other variables remain unchanged. It is usually used in multivariate analysis. When the estimated marginal mean is lower, the value of ΔR is smaller. It can be seen from the figure that the three-dimensional conductive network structure of shape 2 maintains a more stable resistance during stretching.

### 3.2. Simulation and Verification

Therefore, in this paper, the second kind of linear bunch conductive structure is studied in depth. First, two kinds of elastic matrix (Figure 2a) with uniform thickness and equiwall thickness containing linear bunch conductive structures are designed; the enlarged diagram is a schematic diagram of the internal structure of uniform thickness and equiwall thickness. The internal structure in Figure 2a is the bunch structure, while the external structure is of uniform thickness and equiwall thickness. The parameters that determine the elastic matrix of the linear bunch structure are horizontal diameter (a), longitudinal diameter (b), neck length (c), and thickness (d). The Young’s modulus of the elastomer material is 0.4 MPa (Appendix A). This value is used for finite element analysis simulation. Using Yeoh’s mechanical module and electrical module, a finite element analysis and simulation are carried out under the action of fluid–solid coupling. Based on the model of this structure, the change in resistance in the case of stretching is simulated to match the change in resistance in the actual tensile experiment. As shown in Figure 2b, in the simulation process, the linear bunch elastic matrix is stretched and deformed. It can be seen from the figure that the tensile strain is mainly released by the gourd ball with larger curvature, and there is only slight deformation at the neck connection between the balls. The thin neck is indeed the main contributor to the overall resistance. Therefore, using this string structure, it is possible to suppress the change in resistance to a lower level during the stretching process. Next, we perform a finite element simulation and experiment on two sets of samples of uniform thickness and equiwall thickness and compare their results to verify the accuracy of the finite element simulation model. The horizontal diameter (a) of the two groups of samples is 4 mm, the longitudinal diameter (b) is 5 mm, the neck length (c) is 0.75 mm, and the thickness (d) is 1 mm. The resistance of the uniform-thickness and equiwall-thickness bunch structure made by 3D printing is measured under the stretching of the stretching machine. Figure 2c,d show the resistance change curve of the finite element simulation and the actual experiment under 50% stretching. It can be seen from the figure that the resistance change during the simulated stretching process is in good agreement with the actual experimental results, which shows that the finite element simulation model is valid in this study.

In order to grasp the influence of each structural parameter on the change in resistance during stretching, control experiments of the finite element simulation are carried out on them. Figure 2e shows the effect of the change in horizontal diameter (a) on the change in resistance in stretching (50%) when the longitudinal diameter (b) is 5 mm, the neck length (c) is 0.75 mm, and the thickness (d) is 1 mm; the change in resistance decreases with the increase in the horizontal diameter (a). Figure 2f shows that when the horizontal diameter (a) is 3 mm, the neck length (c) is 0.75 mm, and the thickness (d) is 1 mm, the change in longitudinal diameter (b) has an effect on the change in resistance when stretching (50%). The change in resistance decreases first and then increases with the increase in longitudinal diameter (b). There is an optimal value to minimize the change in resistance. Figure 2g shows that when the horizontal diameter (a) is 3 mm, the longitudinal diameter (b) is 5 mm, and the thickness (d) is 1 mm, the change in resistance during stretching (50%) increases with the increase in the neck length (c). Figure 2h shows that when the horizontal diameter (a) is 3 mm, the longitudinal diameter (b) is 5 mm, and the neck length (c) is 0.75 mm, the change in the thickness (d) during stretching (50%) has little effect on the change in resistance, which is basically negligible compared to several other parameters. It can be seen from the diagram that the structure of equiwall thickness is more stable than that of the uniform-thickness structure under the same structural parameters, so when preparing samples for practical application, the linear bunch three-dimensional conductive network structure with equiwall thickness is selected.

### 3.3. Performance Characterization

According to the influence of the parameters of the bunch structure on the resistance change during stretching summarized by the control experiment, a stretchable conductor with basically stable resistance under 50% stretching is further produced by 3D printing. Figure 3a shows the fabrication process of the stretchable conductor, first printing an elastic matrix with a bunch structure of equiwall thickness through a 3D printer, then rinsing the sample in alcohol, removing the support, and filling the channel with liquid metal using the injection method. According to the influence law of the parameters obtained by the finite element simulation, the horizontal diameter (a) is 2.5 mm, the longitudinal diameter (b) is 2 mm, the neck length (c) is 0.25 mm, and the thickness (d) is 0.5 mm. The equiwall-thickness bunch structure has a resistance change of only about 1% when stretching 50%, while the resistance of the ordinary cylindrical pipe changes by about 126% (Figure 3b) under the same stretching condition, which greatly improves the strain stability of the stretchable conductor. In addition, the stability of the bunch structure and the ordinary cylindrical structure during twisting and bending is also tested (Appendix A). It can be seen that when bending to 180 degrees, the resistance of the ordinary cylinder changes by 6.5%, while the resistance of the bunch structure only changes by 0.3%. Similarly, when twisted to 360 degrees, the resistance of the ordinary cylinder changes by 9.3%, while the resistance of the bunch structure changes by only 0.1%. Under different deformation, the bunch structure still maintains excellent stability performance compared with the ordinary cylindrical structure. It indirectly reflects that the design of the equiwall-thickness linear-bunch-structured conductive network plays an important role in obtaining excellent strain insensitive conductivity. The quality factor Q, defined as the percent strain divided by the percent relative resistance change, is an important evaluation factor for the strain-insensitive conductivity of stretchable conductors. As depicted in Table 1, the equiwall-thickness linear bunch structure exhibits a prominent Q value (33) even though it is in a state of strain (50%) compared to recently reported stretchable conductors based on the porous structure [45], wrinkle structure [48], and others [44,47,49,50,51,52,53], suggesting the outstanding strain-insensitive conductivity.

The reason for choosing liquid metal as a filler is that it has very little effect on the mechanical properties of the elastic material, and this low-viscosity liquid easily flows in response to the applied strain [54]. The mechanical properties (Figure 3c) of stretchable conductors with hollow and filled bunch structures are measured by a stretching machine. The bunch structures with and without liquid metal have almost the same mechanical properties, indicating that the effect of liquid metal on mechanical properties is negligible. Under 20 large strain cycles, the change in tensile properties is negligible (Figure 3d). This shows that the elastomer material has a relatively high mechanical durability. In order to further test the stability of the stretchable conductor, Figure 3e shows that we apply different stretching rates (50−200 mm min^−1^), which have little effect on the change in resistance, which is of great significance for the different rates of actual use in different parts of the manipulator. Figure 3f illustrates the change in resistance of the equiwall bunch structure under cyclic extending and relaxing with various strains by exerting the same stimulus rate of 20 mm min^−1^. The influence shown in Figure 3f shows the resistance change apparently under various strain states (10, 30, and 50%), and the response is found to be stable. Finally, in order to evaluate the stability and durability of the equiwall-bunch-structured stretchable conductor, the sample is subjected to 1000 cycles of 50% strain and a 200 mm min^−1^ rate. Figure 3g shows that in 1000 cycles, the resistance change of the sample after cyclic stretching can be neglected relative to the change in the initial resistance. The resistance increase is less than 0.06% after 200 stretching cycles and is less than 0.1% after 1000 stretching cycles, which proves the high dynamic stability of the bunch-structured stretchable conductor. Figure 3h shows that the change in resistance is still stable in 1000 cycles, and the illustration of Figure 3h is the enlarged area in 230−265 s and 3215−3255 s. For most stretchable conductors, repeated stretching often leads to a sharp decline in conductivity due to damage to the internal conductive channel. Due to the good recoverability of the conductive network, this bunch structure can be almost restored to the initial value by releasing the applied strain. It is further shown that this bunch structure has a long working life and reliability.

### 3.4. Principal Analysis

Figure 4 shows the structural change of the designed bunch-structured conductive network before and after stretching. An enlarged view of Figure 4a shows the shape of the neck of the conductor and the ball before and after stretching, and it can be seen that the liquid metal at the ball releases strain and squeezes into the neck during stretching. The ball of this bunch structure is similar to a reservoir, which is supplemented at the neck during stretching, so the resistance is always maintained in a relatively stable state during stretching. Figure 4b shows a schematic diagram of the structural changes of this bunch-structured conductive network and ordinary cylindrical conductive network during stretching, and it is obvious that the ordinary cylindrical conductive network becomes elongated in the middle after stretching, while the neck of the bunch-structured conductive network does not change much [43]. The results of finite element simulations further confirm this conclusion, and when stretched, the neck of the bunch structure resistance will be supplemented by liquid metal from the ball, and even increase the diameter of the neck (Appendix A). Therefore, this bunch-structured stretchable conductor with equiwall thickness has significant durability and stable strain-insensitive performance, and it has great application prospects in flexible bodies and wearable systems.

### 3.5. Application

Now, more and more people use headphones, which can be seen everywhere in our daily lives. While wireless headphones are more widely recognized for their flexibility, cable headphones still have a wide audience because of their music’s conductivity and accuracy. However, the length and breakage of the headphone cord have been troubling users. Therefore, a stretchable headphone line can ensure the stability of signal transmission while maintaining good tensile properties, which can be an optimization solution to this problem. The bunch-structured stretchable conductor designed in this article can be used as a headphone cable with good tensile properties (Figure 5a). To prove its electrical performance, the transmission ability of its musical signal (Chinese national anthem) is tested, which is compared after being unstretched and stretched. By comparing the amplitude data before and after stretching 50% (Figure 5b), we find that the maximum amplitude deviation is only 0.042 V, which hardly affects the quality of the music signal. In addition, the effect of the applied tensile strain on the frequency characteristics of the music signal (Appendix A) is analyzed. The spectrogram is obtained by the Fourier transform of the voltage waveform in Figure 5b. It can be seen that after applying a 50% stretching, its frequency characteristics are similar to those not stretched. In addition, this bunch-like stretchable conductor is used in the charging cable of the mobile phone (Appendix A), and it is found that before and after stretching, it does not affect the charging of the mobile phone at all. Figure 6 directly shows the dynamic stability of the equiwall-bunch-structured stretchable conductor (Appendix A is the cross section of the structure); with a 100 mA current connected, the voltage change after 50% stretching is small, reflecting that the resistance barely changes before and after stretching, and Appendix A shows the real-time change in resistance during dynamic stretching.

## 4. Conclusions

In summary, this paper designs a conductive network with an equiwall-thickness bunch structure, uses 3D printing technology to print an elastic matrix, and then injects liquid metal to form a stretchable conductor. Through experiments and simulations, the influence of the structural parameters of the linear bunch conductive network with equiwall thickness on the resistance change during stretching is summarized, and a stretchable conductor with high dynamic stability is produced according to the summarized rules. When this bunch structure is stretched, the liquid metal at the ball is squeezed, so the neck is replenished, similar to the reservoir structure, which allows the conductor to maintain a stable resistance when stretched. As a stretchable conductor, the resistance changes by only about 1% at 50% tensile strain. In addition, the conductor exhibits high durability and excellent electromechanical stability in 1000 cycles at 50% tensile strain. Due to the high conductivity and superior stretchability of this equiwall-thickness linear bunch conductive network, it has broad application prospects in the field of wearable electronics. It is used in the charging line and headphone line of mobile phones to verify the applicability of the structure as a stretchable wire in practical applications. This work provides inspiration for the development of stretchable conductors based on three-dimensional conductive structures through structural design that can be used in various applications of next-generation wearable devices.

## Figures and Tables

**Figure 1 materials-16-03098-f001:**
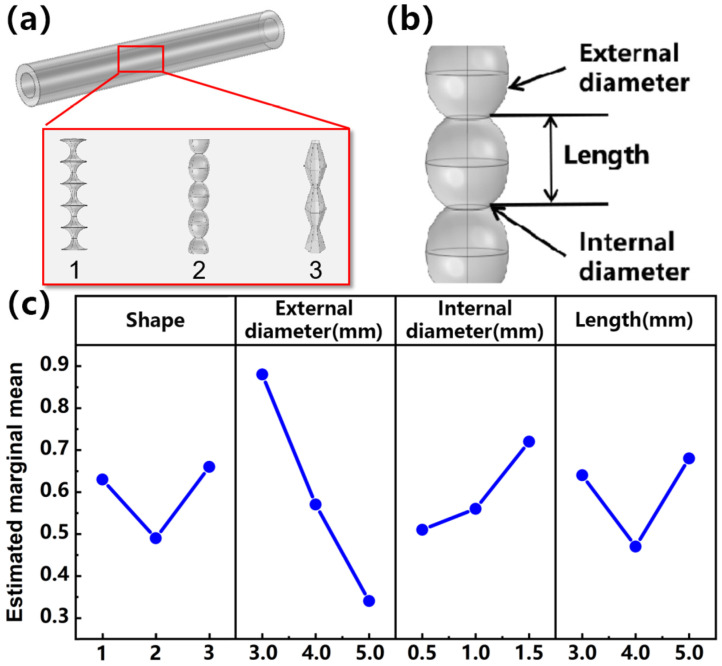
(**a**) Schematic diagram of three structures. (**b**) Structural parameters of the shape. (**c**) Results of orthogonal experiment.

**Figure 2 materials-16-03098-f002:**
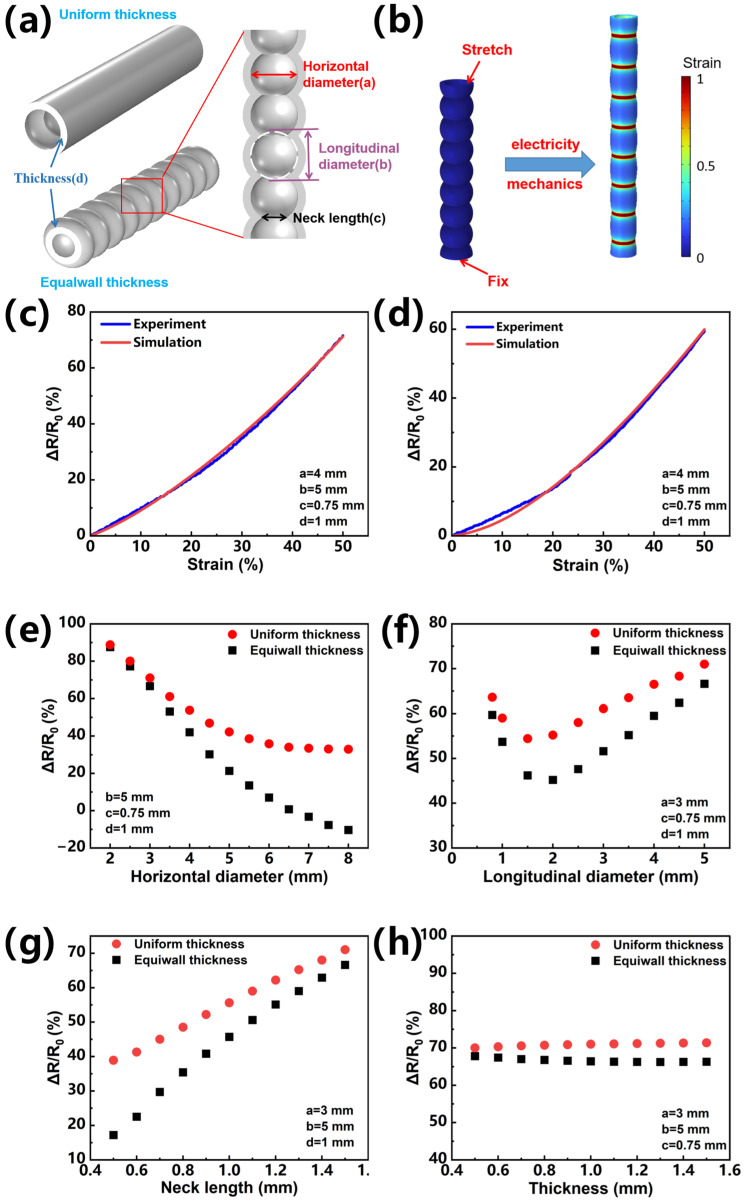
Parameters and electrical properties of 3D-printed bunch structures. (**a**) Schematic diagram of linear bunch conductive network with two elastic matrix shapes. (**b**) Simulation diagram of linear bunch conductive network. Color map represents strain. (**c**) The electrical properties of the experimental and simulated tensile tests of the uniform-thickness bunch structure. a = 4 mm, b = 5 mm, c = 0.75 mm, d = 1 mm. (**d**) The electrical properties of the experimental and simulated tensile tests of the equiwall-thickness bunch structure. a = 4 mm, b = 5 mm, c = 0.75 mm, d = 1 mm. (**e**–**h**) The control experiment of each structural parameter of uniform-thickness bunch structure and equiwall-thickness bunch structure under 50% tensile strain.

**Figure 3 materials-16-03098-f003:**
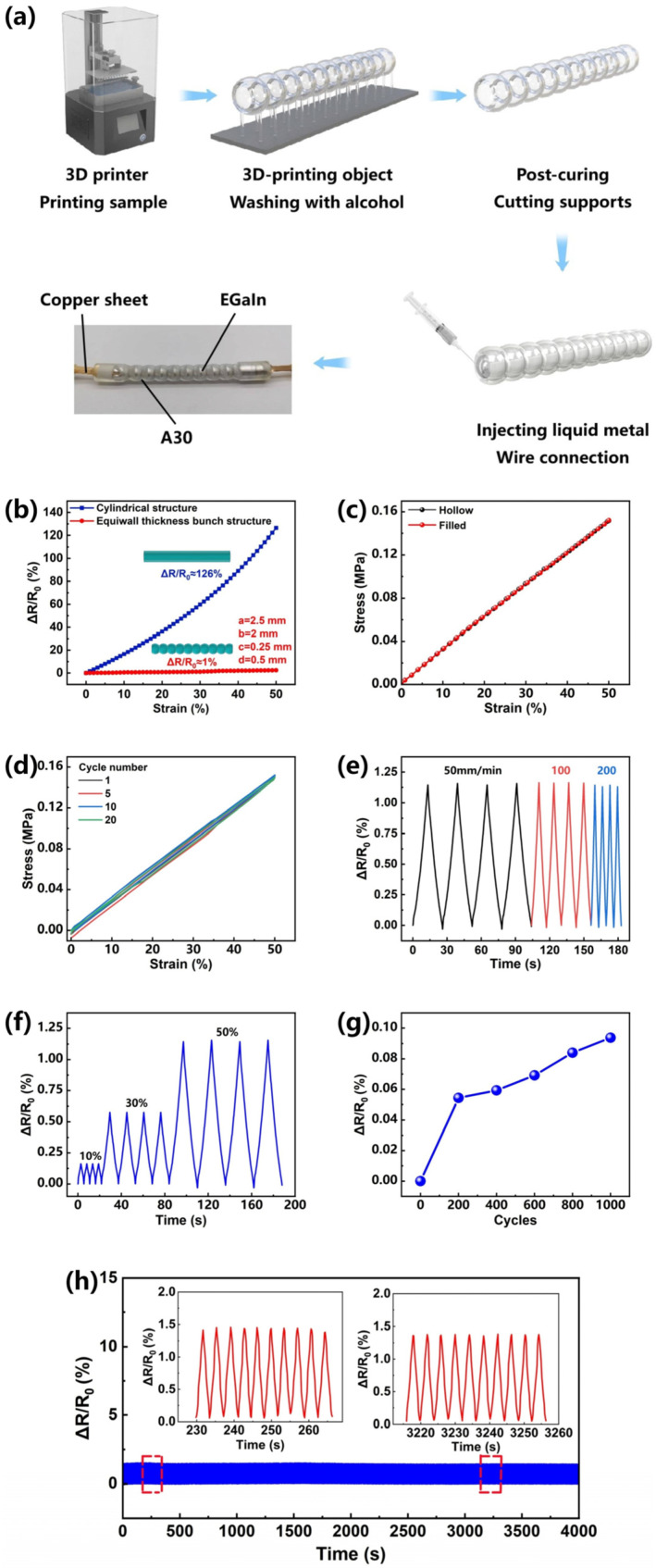
Three-dimensional-printing process and performance of string structure wire. (**a**) Fabrication process. The elastic matrix of the equiwall-thickness linear bunch structure was made by a 3D printer, and the structure was filled with liquid metal. (**b**) Comparison of the resistance change of the strain-insensitive equiwall-thickness bunch structure conductor with the ordinary cylindrical pipe during stretching, through the optimization of structural parameters. (**c**) Comparison of mechanical properties of bunch-structured stretchable conductors filled with liquid metal and unfilled with liquid metal. (**d**) Cyclic diagram of conductor under 50% stretch. (**e**) Variation in resistance with a strain of 50% at different strain rates. (**f**) The resistance change under different strain cycles at a strain rate of 200 mm min^−1^. (**g**) The change in inherent resistance after different stretching cycles. (**h**) Tensile stretching/releasing cycle of 50% strain for 1000 cycles.

**Figure 4 materials-16-03098-f004:**
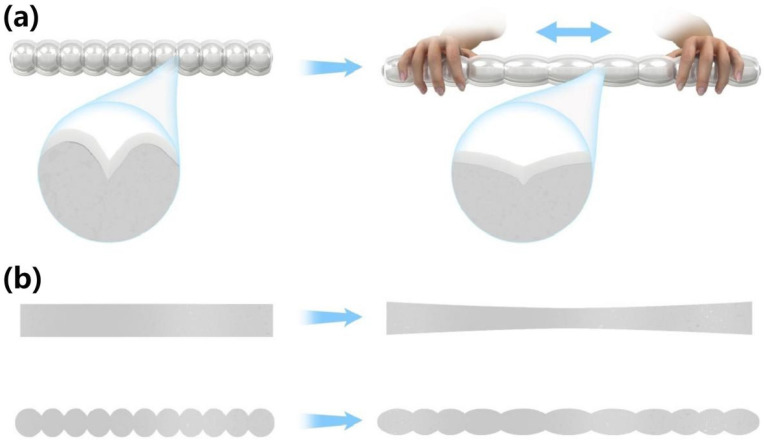
(**a**) Structure diagram of bunch conductive network with equiwall thickness and elastic matrix. (**b**) Schematic diagram of structural changes of equiwall-thickness bunch conductive network and ordinary cylindrical conductive network.

**Figure 5 materials-16-03098-f005:**
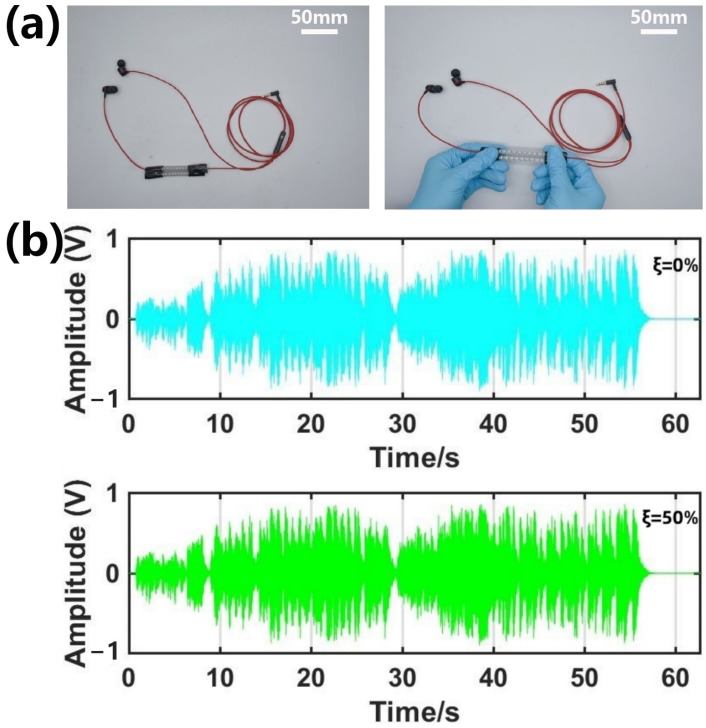
(**a**) Image of stretchable conductor as earphone wire. (**b**) Amplitude–time curve of complex music signals from retractable headphones.

**Figure 6 materials-16-03098-f006:**
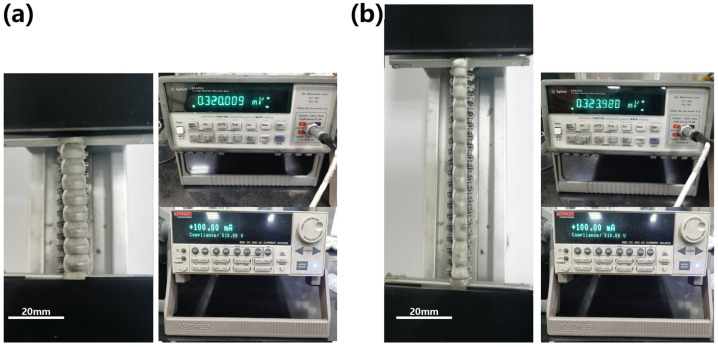
(**a**) Voltage of the initial state of stretchable conductor at 100 mA. (**b**) Voltage of 50% tensile strain of stretchable conductor at 100 mA.

**Table 1 materials-16-03098-t001:** Comparison table of equiwall-thickness bunch structure and other representative structure stretchable conductors.

Conductive Structure	Quality Factor (Q)	Electrical Conductivity	Durability (Number of Cycles)	Reference
welded CNT structure	18 (90% strain)	1.32 S cm^−1^	1000	[44]
porous structure of AgNWs	5 (40% strain)	42 S cm^−1^	100	[45]
wrinkle-crumple rGo structure	23 (70% strain)	/	200	[48]
AgNWs spongy structure	2.5 (50% strain)	27.78 S cm^−1^	1000	[49]
liquid metal corrugated structure	2 (100% strain)	/	500	[50]
AgNWs buckled structure	0.28 (130% strain)	21 S cm^−1^	1000	[51]
micro-wrinkled rGo structure	1.48 (80% strain)	0.256 S cm^−1^	500	[52]
liquid metal spongy structure	25 (50% strain)	478 S cm^−1^	1000	[47]
gold thin-film crack structure	1 (50% strain)	/	500	[53]
liquid metal bunch structure	33 (50% strain)	10,000 S cm^−1^	1000	this work

## Data Availability

The data presented in this study are available on request from the corresponding author.

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
