# Peer review of "Design and 3D Printing of Stretchable Conductor with High Dynamic Stability"

_materials, 2023, doi:10.3390/ma16083098_

Round 1

Reviewer 1 Report

Advantages of paper:

1. Interesting paper showing new possibilities for 3D printing technology in obtaining electroconductive stretchable cables.

2. Good combination of computer modelling and simultanous  testing results in 3D printing technology

Drawbacks of paper:

1. Missing references in Table 1 (last column)

2. Application in robotics - testing only 1000 cycles (last line in caption of Figure 3) is definitely too small.

3. Application in ear phones cables - line 320 ..., the music transmission capacity of stretchable headphones before and after stretching has not changed much, it is almost the same.... is definitely too far. If the Authors claim, that signal is ...almost the same... this must be verified in much better way to reccomend for this application. But on the other hand this is an interesting idea and I will be interested in testing lets say 4 m long ear phone cable.

4. Too numerous irritiating small editorial mistakes easy to correct: eg line 57 should be ...liquid metals [41]....; line 68 - add, that ppy means polypyrrole; line 98 should be ...dynamic stretching.; lines 109, 112 should be 250C, 600C; line 120 should be ...50 mm min-1...; line 125 should be ... 970 kg m-3; line 129 should be ...S-1; line 196 should be ...50%; line 266 should be (50 - 200 mm min-1);

Reviewer 2 Report

This manuscript by Chao Liu et al. reports design and fabrication of stretchable conductor with high dynamic stability through 3D printing technique. Through various experiments and theoretical calculations by finite elemental analysis, they design the stretchable conductor consists of elastic matrix and liquid metal, which exhibited high electrical conductivity and dynamic stability in repeated stretching. Also, they demonstrated two kinds of applications that are headphone cable and mobile phone charging wire. Therefore, in my opinion, this work is suitable for publication in Materials after considering some questions as below:

1. In page 1, line 40, it would be better to change “wearable flexible electronic devices” to “flexible electronic devices” or “wearable electronics devices”.

2. In abstract, it is better to avoid the abbreviation of LM.

3. In page 2, line 68, what is mean the “ppy”? Does “ppy” indicate “polypyrrole”? If it is right, the full name should be added.

4. In page 2, line 89, Does LM indicate liquid metal? It is better that the abbreviation is stated where the full name is firstly mentioned. The authors should carefully concern this issue of other cases.

5. I cannot fully understand the mean of the linear bunch structure. It is better to explain the difference of traditional stretchable conductors and this work in manuscript.

6. The authors should add the cross-sectional SEM or TEM images of developed stretchable conductors, which give microscopic structure.
7. In page 4, line 152, the authors explain about conducting mechanism of liquid metal in stretching states. The proper references should be added, which can support the author’s explanations.

8. In Figure 2, the authors compared the two kinds of elastic matrix. Do two kinds of elastic matrix mean uniform thickness and equiwall thickness? Moreover, Does the uniform thickness in Figure 2e-2h indicate cylindrical structure in Figure 3? The authors should clearly indicate these terms.

9. The authors explained that the developed stretchable conductor had the elongation of 50%. However, in Figure S1, bare elastic matrix showed the elongation of over 100%. Can the developed stretchable conductor retain their property in 100% elongation? If not, please explain why.
